# Polarised moonlight guides nocturnal bull ants home

**Cody A Freas*, Ajay Narenda, Trevor Murray, Ken Cheng**

School of Natural Sciences, Macquarie University, Sydney, Australia

## eLife Assessment

This **important** study substantially advances our understanding of nocturnal animal navigation and the ways that animals use polarized light. The evidence supporting the conclusions is **convincing**, with elegant behavioural experiments in actively navigating ants. The work will be of interest to biologists working on animal navigation or sensory ecology.

**\*For correspondence:**
cody.freas@mq.edu.au

**Competing interest:** The authors declare that no competing interests exist.

## Abstract

For the first time in any animal, we show that nocturnal bull ants use the exceedingly dim polarisation pattern produced by the moon for overnight navigation. The sun or moon can provide directional information via their position; however, they can often be obstructed by clouds, canopy, or the horizon. Despite being hidden, these bodies can still provide compass information through the polarised light pattern they produce/reflect. Sunlight produces polarised light patterns across the overhead sky as it enters the atmosphere, and solar polarised light is a well-known compass cue for navigating animals. Moonlight produces an analogous pattern, albeit a million times dimmer than sunlight. Here, we show evidence that polarised moonlight forms part of the celestial compass of navigating nocturnal ants. Nocturnal bull ants leave their nest at twilight and rely heavily on the overhead solar polarisation pattern to navigate. Yet many foragers return home overnight when the sun cannot guide them. We demonstrate that these bull ants use polarised moonlight to navigate home during the night, by rotating the overhead polarisation pattern above homing ants, who alter their headings in response. Furthermore, these ants can detect this cue throughout the lunar month, even under crescent moons, when polarised light levels are at their lowest. Finally, we show the long-term incorporation of this moonlight pattern into the ants' path integration system throughout the night for homing, as polarised sunlight is incorporated throughout the day.

## Introduction

Many, navigating animals use the position of the sun or moon to guide their movement (*Jander, 1957*; *Klotz and Reid, 1993*; *Perez et al., 1997*; *Dacke et al., 2014*; *Warrant and Dacke, 2016*; *Freas and Cheng, 2022*). Yet these celestial bodies are not always directly visible, often obscured by clouds, the canopy, or after passing below the horizon, resulting in gaps for navigation relying solely upon direct visual detection. Animals hence rely on the pattern of polarised skylight, which is accessible even when the celestial bodies are occluded to some extent (*Horváth et al., 2014*). When the electromagnetic field of light oscillates in a directionally predictable manner, it is defined as polarised. The type of polarisation describes the pattern of this oscillating electromagnetic field. When this oscillation is in a single plane, this light is defined as linearly polarised. The electric field of light can form other patterns, such as circular polarisation, with the field spiralling in three dimensions as a plane wave propagates. Linearly polarised sunlight comprises light waves which occur along a single plane produced as a by-product of light passing through the upper atmosphere (*Horváth and Varjú, 2004*; *Horváth et al., 2014*). The scattering of this light creates an e-vector pattern in the sky,

**eLife digest** Light travels through space as waves, normally oscillating in various orientations. When light waves enter the earth's atmosphere, they become polarised, meaning the waves oscillate along a single plane. Polarised light creates a predictable pattern that is imperceptible to humans but visible to many animals. This pattern provides animals a reliable 'sky compass' based on the sun's position during the day, which is particularly useful because it is visible even when the sun is obscured.

However, many nocturnal animals can successfully navigate well after sunset when this solar sky compass is absent. The moon, reflecting sunlight, creates a similar polarisation pattern in the sky. However, moonlight is about a million times dimmer than sunlight, so nocturnal animals need highly specialised visual systems to detect these lunar cues.

One of these nocturnal animals is the large-eyed bull ant (*Mymecia midas*). These ants rely on solar polarised light to navigate during twilight, when they climb into the canopy of trees surrounding their nest. However, these ants often return home overnight, suggesting they can use other celestial cues when navigating, such as the moon or stars. However, it remains unknown which cues bull ants use at night.

To determine whether bull ants can detect polarised moonlight to orient themselves, Freas et al. used polarisation filters to rotate the polarisation pattern in the night sky above two bull ant nests. The researchers found that when the polarisation pattern was rotated, the forager bull ants altered their direction in line with the change in the polarised pattern, showing that they were using this pattern to navigate. Interestingly, the ants could detect the lunar polarisation pattern even when the moon was below the horizon, or when only a sliver of the moon was reflecting sunlight, indicating extremely high visual sensitivity in these animals.

Freas et al. show for the first time an instance of animals using polarised moonlight patterns to navigate. This type of investigation can be used to develop navigational systems for robotics. The results suggest that a polarisation sensor with sufficient sensitivity can be used to create a sky compass to navigate both at night and during the day.

which is arranged in concentric circles around the sun or moon's position with the maximum degree of polarisation located 90° from the source. Hence when the sun/moon is near the horizon, the pattern of polarised skylight is particularly simple with uniform direction of polarisation approximately parallel to the north-south axes (*Dacke et al., 1999*; *Dacke et al., 2003*; *Reid et al., 2011*; *Zeil et al., 2014*). The pattern's stability makes the sky's polarisation a useful directional cue for orientation (*Wehner and Müller, 2006*; *Reid et al., 2011*; *Lebhardt and Ronacher, 2014*; *Warrant and Dacke, 2016*; *Freas et al., 2017b*; *Freas et al., 2019b*; *Freas and Spetch, 2023*), which insects detect through specialised photoreceptors located in the dorsal rim area of their eyes (*Labhart and Meyer, 1999*; *Homberg and Paech, 2002*; *el Jundi et al., 2015*). Like solar polarisation, though a million times weaker, the moon reflects sunlight, producing a polarised moonlight pattern emanating from the moon's position in the night sky (*Gál et al., 2001*). Given that the moon creates a much dimmer version of the polarisation pattern formed around the sun, only night-navigating insects with eyes highly specialised for low-light detection may be able to rely on this pattern to orient and navigate to goals.

Currently, only dung beetles (the nocturnal *Scarabaeus satyrus* and *S. zambesianus* and the diurnal *Scarabaeus (Kheper) lamarcki*) are known to attend to moonlight polarisation patterns during their movement (*Dacke et al., 2003*; *Dacke et al., 2004*; *Dacke et al., 2011*; *Smolka et al., 2016*; *Foster et al., 2019*). Yet interestingly, these beetles do not use moonlight to navigate, instead relying on this pattern to keep moving straight in order to roll their dung balls expeditiously away from a central dung pile. As such, this cue has only been documented for heading maintenance over short periods. While it has been theorised that this ability to detect the much dimmer polarisation pattern produced by the moon may be present across nocturnal insects more broadly, including nocturnal bees and crickets (*Herzmann and Labhart, 1989*; *Greiner et al., 2007*; *Rost and Honegger, 1987*), there is currently no behavioural evidence for its use in goal-directed navigation. We sought such evidence in nocturnal bull ants.

The large-eyed *Myrmecia* ants have several species that restrict the majority of their navigation to evening twilight (outbound) and morning twilight (inbound), respectively (*Narendra et al., 2017*). We

know that two well-studied nocturnal ant species, *Myrmecia pyriformis,* and *Myrmecia midas*, use the overhead solar polarised light pattern, which is still visible during the twilight period to derive compass information (*Reid et al., 2011*; *Freas et al., 2017b*; *Freas et al., 2017c*; *Freas et al., 2018*). Because the information required for visual navigation degrades beyond twilight, it has been suggested that animals tend to be less active at night. However, a small proportion of *M. pyriformis* foragers leave the nest (10.7% of daily foraging force) or return home during the night (13.3% of daily foraging force) (*Reid et al., 2013*). In *M. midas*, this nocturnal activity is even more pronounced with the majority of the foraging force returning during the night (62.8% of daily foraging force) along with a minority of foragers leaving the nest during the night (26.2% of daily foraging force) (*Freas et al., 2017b*).

Nocturnal bull ants navigate using a combination of learned visual cues (*Freas et al., 2018*; *Islam et al., 2020*; *Deeti et al., 2023*) and homing vectors obtained by integrating pedometric and celestial compass information (*Wehner and Srinivasan, 2003Wittlinger et al., 2006*; *Wehner, 2020*). This 'true nocturnal navigation' is likely aided by the increase in the light intensity provided by the moon's presence. In *M. pyriformis*, on 'full-moon' nights there was a significantly greater proportion of foragers leaving the nest at night compared to a 'new-moon' night (*Reid et al., 2013*). This additional light at night may enhance terrestrial visual features foragers have learned and, therefore, assist in visual guidance. In addition, the moon and the lunar polarisation pattern also likely provide compass information, allowing foragers to acquire or follow a homing vector. Several arthropods, including ants and bees, directly track the moon's position to obtain compass information (*Jander, 1957*; *Klotz and Reid, 1993*; *Dacke et al., 2004*; *Ugolini et al., 2013*). But given the moon may be occluded by clouds or overhanging canopy we aim to identify here whether the lunar polarised skylight can also be used by ants for homing.

In the current study, we studied foraging ants' ability to orient during lunar twilight, by placing and rotating a linear polarising filter over them as they returned to the nest (*Figure 1A*). This rotation blocks the ambient e-vector direction of the sky above the navigator, replacing it with an artificial e-vector of polarisation that is rotated ±45° off the ambient pattern. We also explored whether these navigators weigh their attendance to polarised moonlight across the lunar cycle, since during quarter-moon and crescent-moon nights, smaller portions of the moon's surface reflect sunlight (as well as moonless nights when no ambient e-vector is present). Finally, we characterised changes in weighting that these ant navigators give polarised moonlight, as a function of the moon's overnight consistency in the sky (waxing vs. waning) and the length of their accumulated path integrator, which should increase the weight given to celestial compass portion of the path integrator when in conflict with terrestrial visual cues (*Burkhalter, 1972*; *Narendra, 2007*; *Wystrach et al., 2015*; *Freas and Cheng, 2019a*).

## Results

### Full moon testing

Under a (waxing) *Full Moon* with lunar illumination above 80%, when the linear polariser was rotated clockwise (+45°), exit orientations were shifted to the right of initial headings (mean ± s.e.m. Nest 1: 38.4 ± 4.8°; Nest 2: 23.4 ± 4.2°; *Figure 2A and B*), and these changes were significant (Moore's Paired Test, Nest 1: $R$=1.639, p<0.001; Nest 2: $R$=1.592, p<0.001). Forager headings also changed predictably when the filter was rotated counter-clockwise (−45°) with exit orientations to the left of initial headings (Nest 1: −41.1 ± 5.5°; Nest 2: −27.1 ± 7.4°; *Figure 2A and B*), and these changes were significant (Moore's Paired Test, Nest 1: $R$=1.794, p<0.001; Nest 2: $R$=1.310, p<0.01). After exiting the +45° or −45° rotated filter, foragers reoriented significantly to the left (Moore's Paired Test, Nest 1: $R$=1.598, p<0.001; Nest 2: $R$=1.383, p<0.005) or right, respectively (Moore's Paired Test, Nest 1: $R$=1.604, p<0.001; Nest 2: $R$=1.328, p<0.005). Shift magnitudes did not differ between +45° and −45° conditions (Watson–Williams F-test, Nest 1: $F_{(1,23)}$=0.155, p=0.697; Nest 2: $F_{(1,20)}$=0.234, p=0.634). Shift magnitudes were significantly larger at Nest 1 compared to Nest 2 (Watson–Williams F-test, $F_{(1,45)}$=8.672, p=0.005), exhibiting shifts near the full e-vector change (39.8 ± 3.2°) while foragers at Nest 2 only exhibited shift magnitudes at about half the 45° rotation (25.2 ± 3.7°). A Var test showed no significant difference in circular variance between nests (Var Test; $Z$=−0.09; p=0.530).

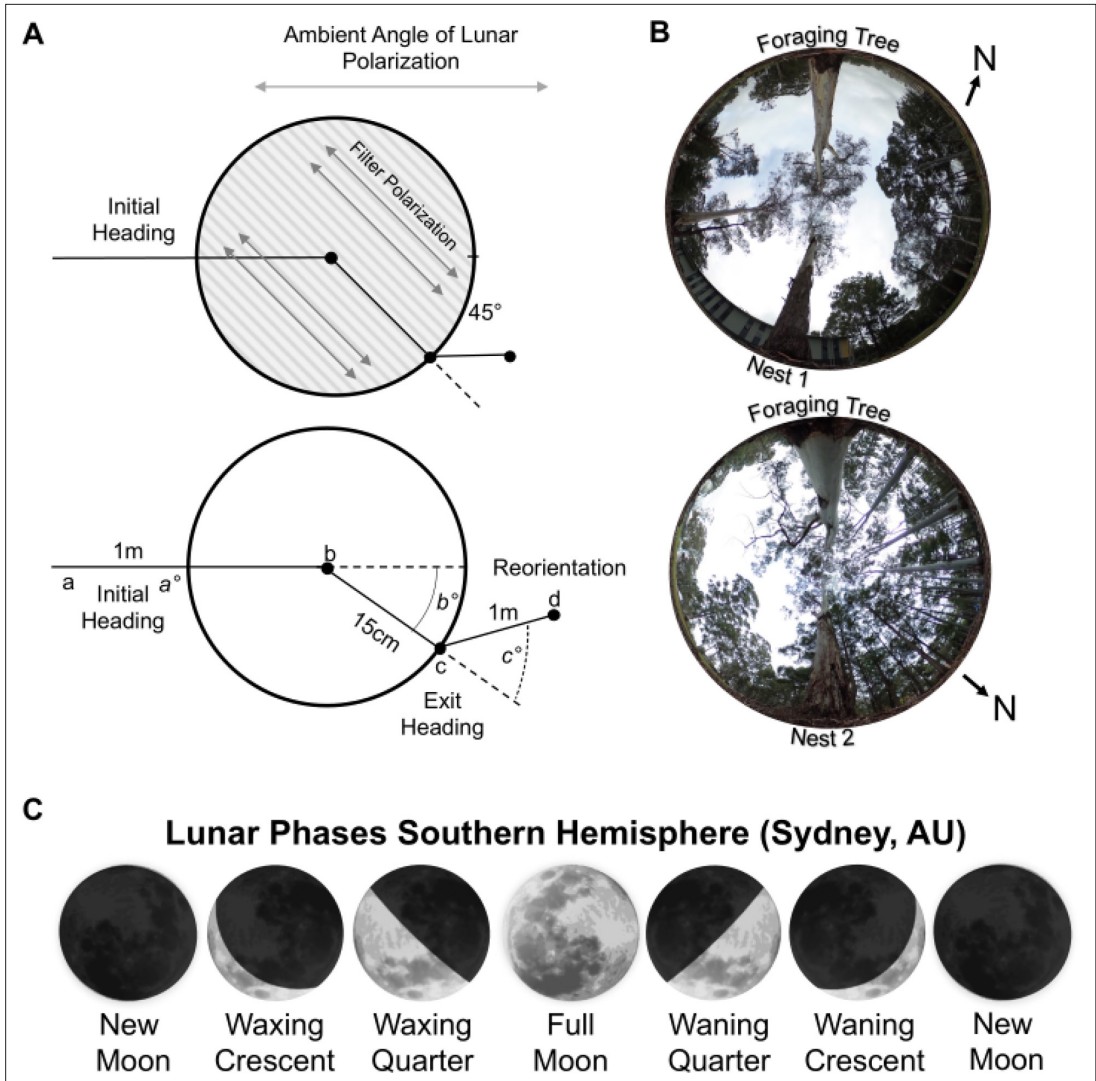

**Figure 1.** Diagram of the polarisation filter and changes to the overhead lunar polarisation pattern. (**A**) During the inbound journey, a linear polarisation filter was placed over the forager, rotating the overhead e-vector by ±45°. Panel depicts the positional measurements recorded during testing. Initial orientation routes were measured from the foraging tree release point (a) to when the polarisation filter was placed over each forager (b). Exit orientations were measured from the filter centre (b) to the forager's exit location at the filter edge (c). Route directions under the filter (b°) were calculated from the forager's initial route direction zeroed. Reorientations were measured from the forager's exit location from the polarisation filter (c) to the forager's path 1 m after exit (d). Reorientation directions (c°) were calculated from the under-filter route direction zeroed. (**B**) Images of the sky and canopy cover at both nests. Photos were taken at the on-route midpoint between the foraging and nest trees. Both Nests were located on the edge of tree stands. However, Nest 1 was in a more barren area of the field site, with less vegetation both along the horizontal plane and in the overhead canopy. (**C**) Lunar phases denote the sunlit part of the moon's surface and where this area is increasing (Waxing) or decreasing (Waning). The lunar phase cycle repeats every 29.5 d. Moon images are public domain art accessed through Wiki Commons (https://commons.wikimedia.org/).

## Waxing lunar phases

Under a *Waxing Quarter Moon* (lunar illumination ~50%), when the e-vector was rotated clockwise (+45°), exit orientations were again significantly shifted (Moore's Paired Test, $R$=1.787, p<0.001) to the right of their initial heading (mean ±s.e.m.; 37.6 ± 4.1°; *Figure 3A*). Forager headings were similarly significantly altered (Moore's Paired Test, $R$=1.734, p<0.001) when the overhead e-vector was rotated counter-clockwise (–45°) with exit orientations to the left of initial headings (mean ±s.e.m. –38.5 ± 6.4°; *Figure 3A*). After exiting the filter in both conditions, foragers reoriented significantly back to the ambient lunar e-vector either to the left (+45°: Moore's Paired Test, Nest 1: $R$=1.616, p<0.001) or right (–45°: Moore's Paired Test, Nest 1: $R$=1.664, p<0.001) of their filter exit heading direction.

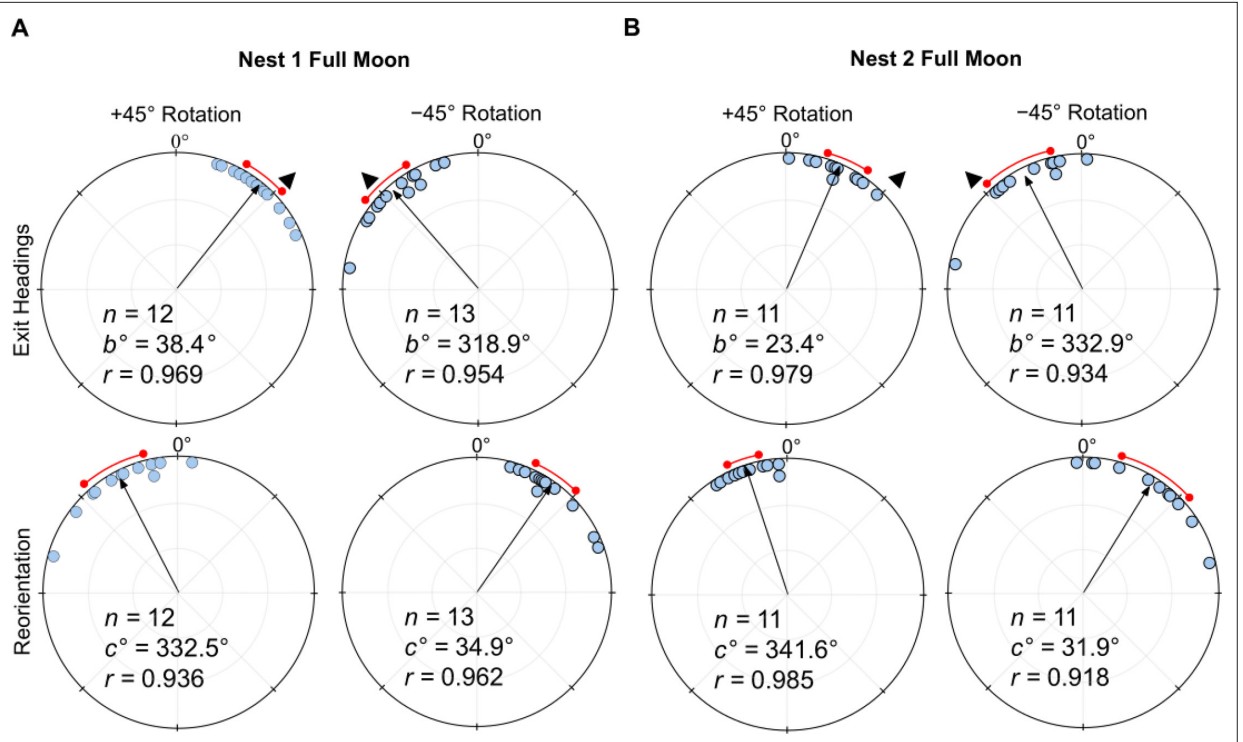

**Figure 2.** Circular distributions of headings during the original full moon conditions. In both conditions, testing occurs on the nights preceding the full moon (illumination >80%) with the moon waxing. Circular plot shifts show the exit orientations of individual foragers relative to initial headings while the reorientation represents the change in headings 1 m after exiting the filter. Triangles denote the ±45° e-vector rotation. The arrow denotes the length and direction of the mean vector while red arc denotes the 95% confidence interval of headings. (**A**) Nest 1 foragers, 5 m from the nest (6.0 m foraging route). (**B**) Nest 2 foragers, 2 m from the nest (3.1 m foraging route). n, number of individuals; b°, mean vector of shift; c°, mean vector of reorientation; r, length of the mean vector.

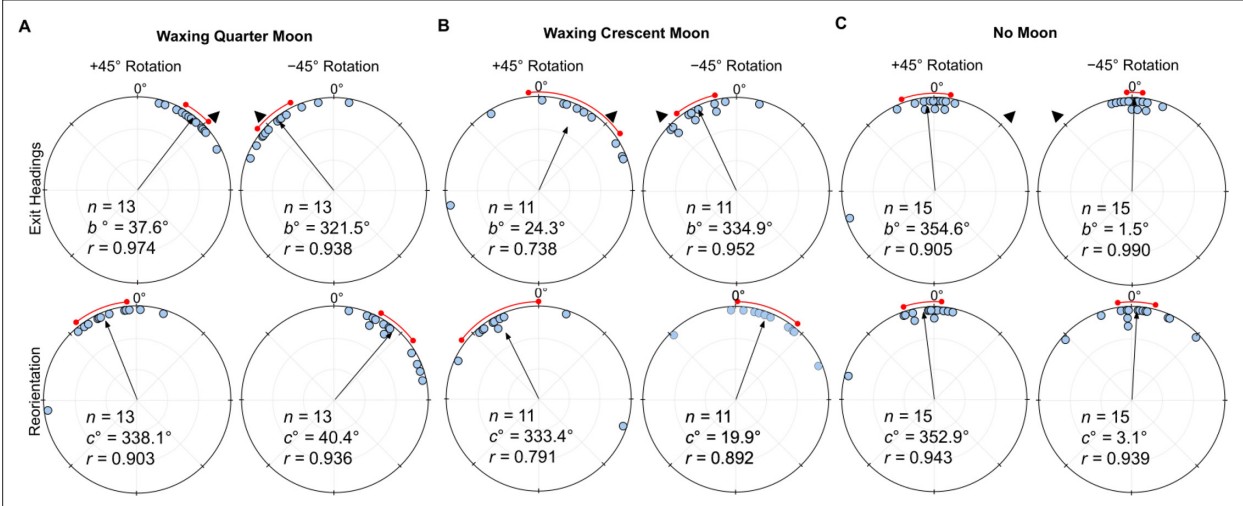

**Figure 3.** Circular distributions of headings during (**A**) Waxing *Quarter Moon,* (**B**) *Waxing Crescent Moon,* and (**C**) *No Moon* conditions. Circular plot shifts show the exit orientations of individual foragers relative to initial headings while the reorientation represents the change in headings 1 m after exiting the filter. Triangles denote ±45° e-vector rotation. The arrow denotes the length/direction of the mean vector. n, number of individuals; b°, mean vector of shift; c°, mean vector of reorientation; r, length of the mea vector. The red arc denotes the 95% convedence interval of headings.

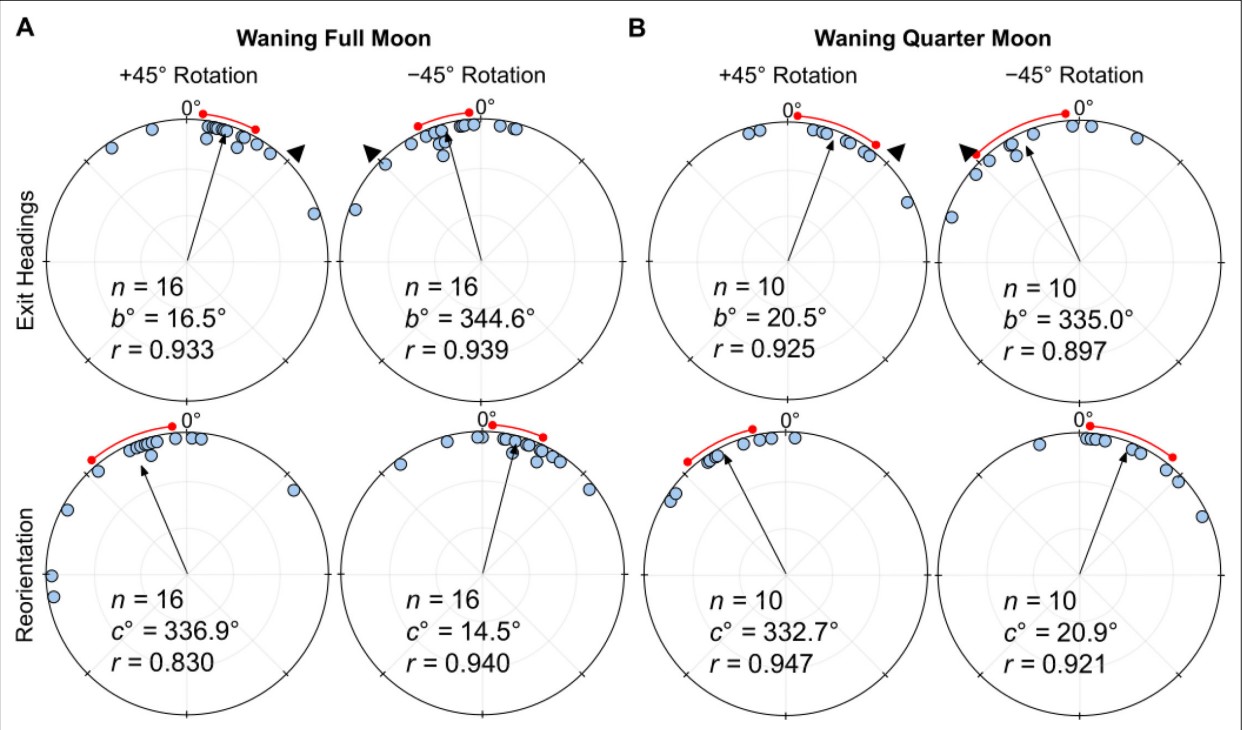

**Figure 4.** Circular distributions of headings during (**A**) Waning Full *Moon,* (**B**) *Waning Quarter Moon* conditions. Circular plot shifts show the exit orientations of individual foragers relative to initial headings while the reorientation represents the change in headings 1 m after exiting the filter. Triangles denote ±45° e-vector rotation. The arrow denotes the length/direction of the mean vector while the red arc shows the 95% confidence interval of headings. *n*, number of individuals; *b°*, mean vector of shift; *c°*, mean vector of reorientation; *r*, length of the mean vector.

Foragers continued to show evidence of attending to the lunar polarisation pattern even under a *Waxing Crescent Moon* (lunar illumination ~20%). Here, when the e-vector was rotated clockwise (+45°), exit orientations were again significantly shifted (Moore's Paired Test, $R=1.175$, $p<0.025$) to the right of their initial heading (mean ±s.e.m.; 24.3 ± 15.5°; *Figure 3B*). Forager headings were similarly significantly altered (Moore's Paired Test, $R=1.39$, $p<0.005$) when the overhead e-vector was rotated –45° counter-clockwise, with exit orientations to the left of initial headings (mean ±s.e.m.; –25.1 ± 6.3°; *Figure 3B*). After exiting the filter in both conditions, foragers reoriented significantly back toward the ambient lunar e-vector either to the left (+45°: Moore's Paired Test, Nest 1: $R=1.324$, $p<0.005$) or right (–45°: Moore's Paired Test, Nest 1: $R=1.223$, $p<0.025$) of their filter exit heading direction. Shift magnitudes were not significantly different between ±45° conditions (Watson–Williams F-test, $F_{(1,20)}=0.03$, $p=0.863$).

When no ambient lunar e-vector was present (*No Moon*) and the polariser was rotated either clockwise (+45°) or counter-clockwise (–45°), foragers did not significantly alter their paths under the filter (+45°: Moore's Paired Test, $R=0.226$, $p>0.50$; mean ±s.e.m.: –5.4 ± 7.4°; –45°: Moore's Paired Test, $R=0.650$, $p>0.10$; mean ±s.e.m.: 1.5 ± 2.4°; *Figure 3C*). Foragers also did not significantly reorient after exiting the filter (+45°: Moore's Paired Test, $R=0.294$, $p>0.50$; mean ±s.e.m.: –7.1 ± 5.6°; –45°: Moore's Paired Test, $R=0.611$, $p>0.10$; mean ±s.e.m.: 3.1 ± 5.8°). Shift magnitudes were not significantly different between ±45° conditions (Watson–Williams F-test, $F_{(1,24)}=0.016$, $p=0.899$) .

When comparing shift magnitudes across lunar phases, *Full Moon* foragers were not significantly different from either the *Waxing Quarter Moon* or *Waxing Crescent Moon* (Watson–Williams F-test, $p>0.05$).

## Waning lunar phases

Under a *Waning Full Moon*, when the polarisation e-vector was rotated clockwise (+45°), exit orientations were shifted significantly to the right of initial headings (mean ±s.e.m.=16.5 ± 5.3°; Moore's Paired Test, $R=1.468$, $p<0.005$; *Figure 4A*). Headings also shifted significantly to the left of initial

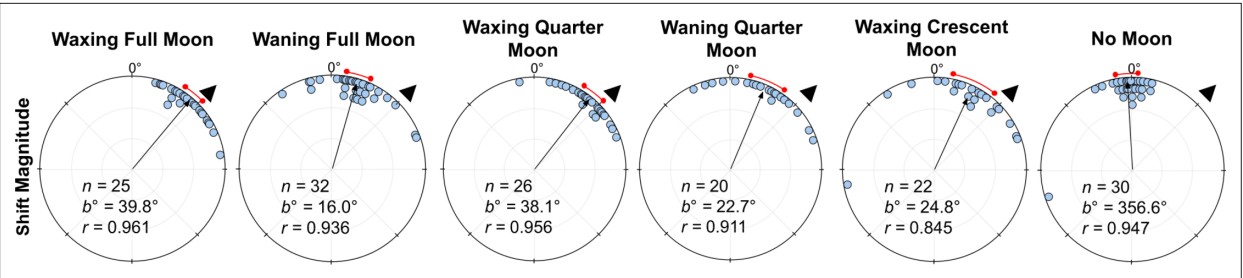

**Figure 5.** Shift magnitudes for lunar phase conditions at Nest 1. Each circular plot shows the ±45° combined shifts for each condition. Triangles denote +45° e-vector rotation; data from −45° were mirrored and combined. The arrow denotes the length and direction of the mean vector. *n*, number of individuals; *b*°, mean vector of shift; *c*°, mean vector of reorientation; *r*, length of the mean vector. The red arc denotes the 95% confidence interval of headings.

headings (mean ±s.e.m.: −15.4 ± 5.1°; Moore's Paired Test, $R=1.513$, p<0.001; *Figure 4A*) when the overhead e-vector was rotated −45° counter-clockwise. After exiting the filter in both conditions, foragers reoriented significantly back to the ambient lunar e-vector either to the left (+45°: Moore's Paired Test, Nest 1: $R=1.548$, p<0.001) or right (−45°: Moore's Paired Test, Nest 1: $R=1.247$, p<0.025) of their filter exit heading direction. Shift magnitudes were not significantly different between the ±45° conditions (Watson–Williams F-test, $F_{(1,30)}=0.022$, p=0.884).

Under a *Waning Quarter Moon* (lunar illumination ~50%), when the e-vector was rotated clockwise (+45°), exit orientations were shifted to the right of initial headings (mean ±s.e.m.=20.5 ± 8.4°; *Figure 4B*), and these changes were significant (Moore's Paired Test, $R=1.33$, p<0.005). Headings also changed significantly (Moore's Paired Test, $R=1.31$, p<0.01) when the overhead e-vector was rotated counter-clockwise (−45°) with exit orientations to the left of initial headings (mean ±s.e.m.: −25.0 ± 9.9°; *Figure 4B*). After exiting the filter in both conditions, foragers reoriented significantly back to the ambient lunar e-vector either to the left (+45°: Moore's Paired Test, Nest 1: $R=1.504$, p<0.001) or right (−45°: Moore's Paired Test, Nest 1: $R=1.246$, p<0.025) of their filter exit heading direction. Shift magnitudes were not significantly different between the ±45° conditions (Watson–Williams F-test, $F_{(1,18)}=0.154$, p=0.700).

When comparing shift magnitudes between Waxing and Waning phases, shift magnitude was significantly higher in both *Waxing Full Moon* and *Waxing Quarter Moon* foragers when compared to their *Waning* counterparts (39.8° vs. 16.0° and 38.1° vs. 22.7°, respectively; Full Moon: Watson–Williams F-test, $F_{(1,55)}=21.62$, p<0.001; Quarter Moon: Watson–Williams F-test, $F_{(1,44)}=5.889$, p=0.038; *Figure 5*). Var tests showed no significant difference in circular variance between Waxing and Wanning phase shift magnitudes (Full Moon: Var Test; $Z=−0.50$; p=0.625; Quarter Moon: Var Test; $Z=−0.38$; p=0.716).

## Vector testing

Both forgers with a long ~6 m remaining vector (*Halfway Release*), or a short ~2 m remaining vector (*Halfway Collection & Release*), tested at the same location, exhibited significant shifts (*Halfway Release*: Moore's Paired Test, $R=1.728$, p<0.001; *Halfway Collection & Release*: Moore's Paired Test, $R=1.380$, p<0.005) to the right of initial headings (mean ±s.e.m.: 41.9 ± 4.9° and 16.8 ± 4.3°; *Figure 6B and C*) when the e-vector was rotated clockwise (+45°). Forager headings also significantly shifted (*Halfway Release*: Moore's Paired Test, $R=1.664$, p<0.001; *Halfway Collection & Release*: Moore's Paired Test, $R=1.07$, p<0.05) when the overhead e-vector was rotated counter-clockwise (−45°) with exit orientations to the left of initial headings (mean ±s.e.m.: −46.4 ± 6.3° and −12.7 ± 8.5°; *Figure 6B and C*). After exiting the filter in both conditions, foragers reoriented significantly back to the ambient lunar e-vector either to the left (+45°: *Halfway Release*: Moore's Paired Test, $R=1.692$, p<0.001; *Halfway Collection & Release*: Moore's Paired Test, $R=1.600$, p<0.001) or right (−45°: *Halfway Release*: Moore's Paired Test, $R=1.604$, p<0.001; *Halfway Collection & Release*: Moore's Paired Test, $R=1.274$, p<0.01) of their exit heading direction. Shift magnitude was significantly higher in *Halfway Release* foragers compared to *Halfway Collection & Release* foragers tested at the same location 2 m from the nest entrance (44.1° and 14.8°, respectively; Watson–Williams F-test, $F_{(1,40)}=29.105$, p<0.001; *Figure 6B*

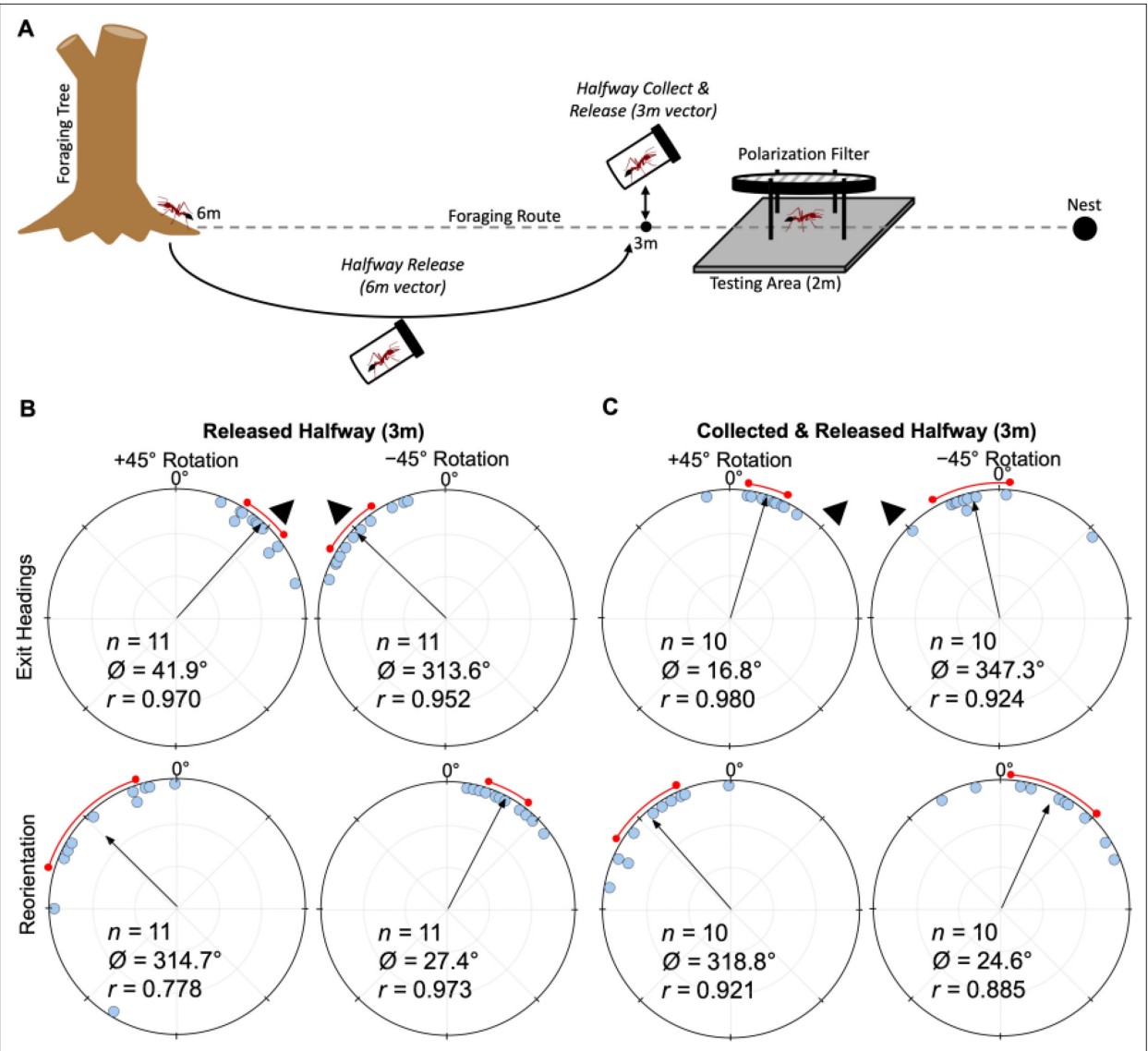

**Figure 6.** Diagram of collection procedures and circular distributions of *M. midas* headings during *Released Halfway, Collected & Released Halfway* conditions. (**A**) Foragers in both conditions were tested at 2 m from the nest with (**B**) *Released Halfway* foragers having a long 6.0 m vector and (**C**) *Collected & Released Halfway* foragers having a 3.1 m vector. Circular plot shifts show the exit orientations of individual foragers relative to their initial headings while the reorientation represents the change in headings 1 m after exiting the filter. Triangles denote ±45° e-vector rotation. The arrow denotes the length and direction of the mean vector. *n*, number of individuals; *b*°, mean vector of shift; *c*°, mean vector of reorientation; *r*, length of the mean vector. The red arc denotes the 95% confidence interval of headings.

*and C*). Var tests showed no significant difference in the circular variance of shift magnitudes when foragers had a long or short vector (Var Test; *Z*=–1.34; *p*=0.188).

## Discussion

These results constitute the first instance of polarised moonlight use for homing and only the second reported instance of its use for orientation in any animal. *Myrmecia midas* foragers predictably altered their heading directions in response to experimental rotations in the overhead lunar polarisation pattern. This ability to detect and use of polarised moonlight persisted throughout the lunar cycle, with foragers attending to the pattern even under a crescent moon with ~20% lunar illumination. This indicates that polarised moonlight is detectable across the lunar month, making it a stable cue these ants can use when moving or updating their path integrator overnight.

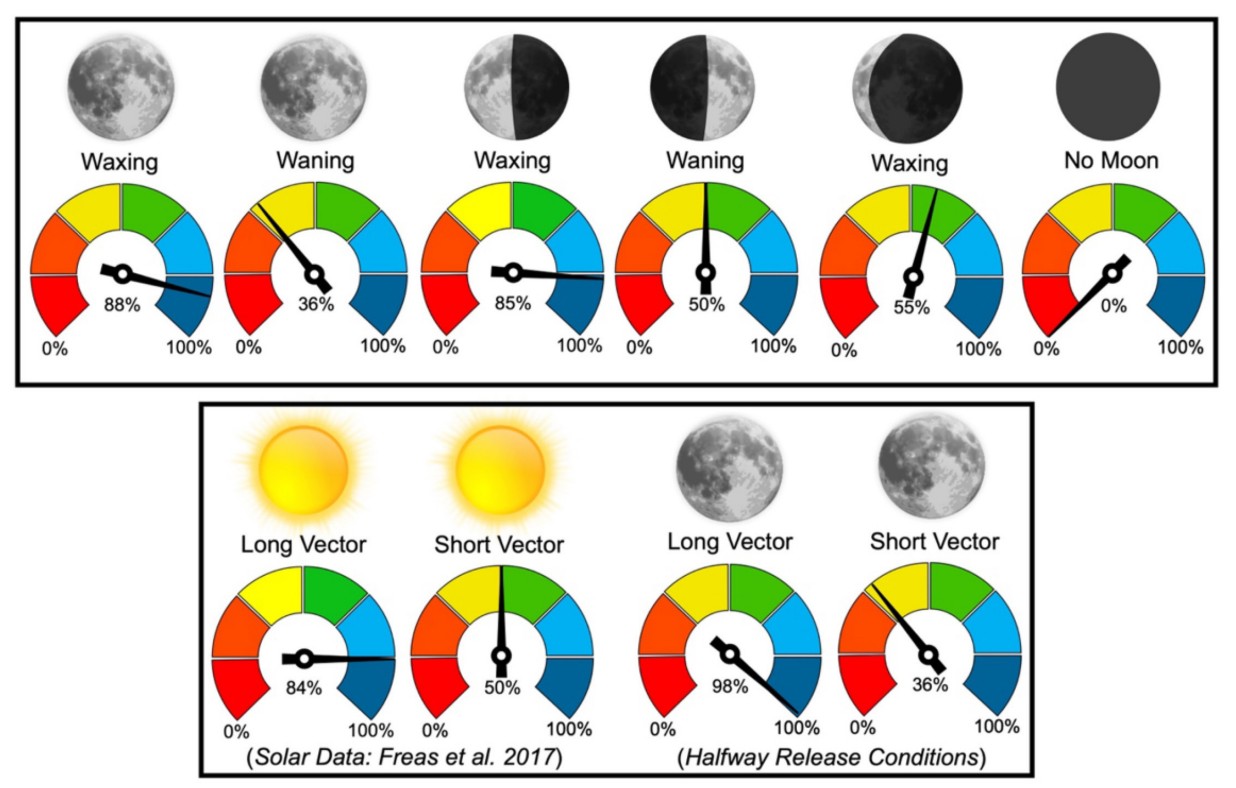

**Figure 7.** Mean shift magnitudes, +45° and –45° combined, were reported as percentages of the 45° e-vector rotation (100%=45°). Vector data for solar polarised light is reported by *Freas et al., 2017b*; *Freas et al., 2017c*, Royal Society Open Science. Sun and moon images are public domain art accessed through Wiki Commons (https://commons.wikimedia.org/).

While they can utilise the lunar polarisation pattern through the lunar cycle, foragers exhibited reduced heading shifts during the waning lunar phases, during which the moon's absence for a portion of the night leads to a cue gap overnight (*Figure 7*). These reductions in observed shifts are likely due to either decreased weighting or a degradation in the celestial compass due to periods when no celestial cues are available. The shift magnitude differences between conditions also point to moonlight polarisation being continuously tracked throughout the overnight period, in line with this celestial cue being integrated into the navigator's path integrator. Further evidence of this cue's integration into the path integrator is illustrated in our vector-length testing conditions. Here, foragers with longer home vectors under a full moon responded almost fully to e-vector changes regardless of their distance to the nest, while foragers with shorter vector distances led to less than half the e-vector shift, corresponding with decreases in vector cue strength at small distances (*Wystrach et al., 2015*). These changes align with how *M. midas* uses polarised sunlight as part of its vector-based homing during evening and morning twilight (*Freas et al., 2017c*; *Figure 7*), suggesting that polarised moonlight is detected and integrated into its path integrator along the same visual pathways as sunlight.

## Moonlight vs. sunlight

The nocturnal bull ants *Myrmecia pyriformis* and *Myrmecia midas* are known to use the solar-polarised light pattern during the twilight periods (*Reid et al., 2011*; *Freas and Cheng, 2017a*; *Freas et al., 2017b*), yet both species are active after twilight, when solar polarisation cues are absent (*Reid et al., 2011*; *Reid et al., 2013*; *Freas et al., 2017b*; *Freas et al., 2017c*). The observed true nocturnal navigation in these animals could be driven by the moon's presence with *M. pyriformis*, which shows more foraging activity on full moon nights (*Reid et al., 2013*). *M. midas* exhibits a high level of overnight activity, with almost half of the foragers returning before the morning solar twilight. *M. midas* also navigates through heavily canopied forest habitat where the moon may be occluded but its

polarisation pattern across the sky remains unobstructed. Thus, *M. midas* makes for an interesting species to characterise lunar and solar polarised light detection.

While we cannot compare solar and moonlight polarisation navigation in outbound ants (outbound foraging is highly correlated with evening twilight when the solar light would overpower any moonlight polarisation pattern), striking similarities occur when comparing solar and moonlight polarisation navigation in ants homing to the nest (*Figure 7*). In the solar polarisation study (*Freas et al., 2017c*), inbound foragers tested during morning twilight at 4–6 m from their nest altered their paths under the filter for almost the full 45° solar e-vector manipulation (–45° rotation: –41.16°/+45° rotation: 34.13°; *Figure 7*) but only compensated for around half of the rotation when tested 1–2 m from the nest (–45° rotation: –24.86°/+45° rotation: 19.73°). We see the same pattern with polarised moonlight with foragers exhibiting near full shifts 5 m from Nest 1 (–45° rotation: –41.4°/+45° rotation: 38.4°) and half shifts 2 m from Nest 2 (–45° rotation: –27.1°/+45° rotation: 23.4°; *Figure 7*).

We also observed consistent slight under-estimation in the shifts even when foragers had a long vector. Observations of ants after the filter was placed overhead suggest that heading updates are not immediate, occurring only after the ant travels along its original heading a few centimetres (~5 cm). This means that even if the ant fully shifts its heading, the delay will cause our measurements at the filter exit to slightly underestimate each individual's position since we measure the angle from where the lunar cue changes (as the filter was placed overhead) and not the position at which the ant altered its heading.

## Moonlight and the path integrator

We also see the same full heading changes to both solar- and moonlight-polarisation-pattern rotations when foragers were released halfway to the nest and tested 2 m with a longer 6 m vector (Solar: –45° rotation: –35.77°/+45° rotation: 39.42°; Moonlight: –45° rotation: –46.4°/+45° rotation: 41.9; *Figure 7*). In the *Halfway collect & Release* condition, when the vector length was 2 m, this shift significantly decreased at the same testing location, indicating that the vector length and not the testing site dictated this adjustment in cue weighting. These findings suggest that under the filter, foragers use any available celestial and terrestrial cues that are still visible, yet the weighting of the polarisation pattern appears to change in accordance with the vector state (*Burkhalter, 1972*; *Narendra, 2007*) and not its test location close to the nest tree, which could be a potentially highly salient landmark. This leads to several interesting implications. First, these ants weight the polarisation cue more highly and perform larger heading shifts when their current path integrator distance is longer. These increased shift magnitudes align with the hypothesis that with longer accumulated vectors, ants increase the weighting given this cue (*Burkhalter, 1972*; *Narendra, 2007*; *Wystrach et al., 2015*; *Freas et al., 2017c*). Second, these ants are using polarised moonlight precisely the same way they use solar polarisation, meaning polarised moonlight is likely integrated into the forager's path integrator throughout the night. Polarised moonlight is likely processed through the same visual pathways as polarised sunlight, meaning that these ants can use the same underlying neural architecture for polarised solar light and polarised moonlight cues. The limiting factors to lunar cue use for navigation would instead be the ant's detection threshold for either absolute light intensity, polarization sensitivity, or spectral sensitivity. In addition to being dimmer, moonlight is less UV-rich compared to direct sunlight, and its spectrum changes across the lunar cycle (*Palmer and Johnsen, 2015*), with sensitivity to green spectrum within polarised moonlight of potential importance for orientation (*Yilmaz et al., 2024*).

## Polarised moonlight and lunar phase

Foragers showed clear evidence of detecting and employing polarised moonlight when homing to the nest across the lunar cycle, even on waxing crescent moon nights. This aligns with polarised moonlight's use in dung beetles, with individuals able to maintain their straight-line paths under quarter and crescent moon e-vectors (*Dacke et al., 2004*). Furthermore, the lack of a shift-magnitude reduction between full and crescent nights suggests no reduction in detection. While we could have continued to test with smaller portions of the moon's surface illuminated, a reduction in shift magnitude could result from either physiological limits in detection or 'decisional' processes in how much weight to accord the cue. Behavioural responses and physiological limits cannot be untangled behaviourally, and detection thresholds would require intracellular recordings under dim polarised light. Finally, the

lack of shifts with *No Moon* foragers indicates that these navigators do not fall back on memories of the evening or morning solar e-vector when presented one overnight.

One unexpected finding was the reduction in shift magnitude under waning moons relative to waxing moons. When we first tested foragers under *Waning Full Moons* and Waning *Quarter Moons* they showed clear evidence of attending to the polarisation cue, but shift magnitudes were significantly smaller compared to the *Waxing Moon* conditions. This reduction in shift magnitude suggests that polarised moonlight was being detected but it was weighted weakly, perhaps due to the incorporation of the celestial compass into the path integrator being interrupted. This observed reduction in shift magnitude likely results from the period overnight in which the waning moon was absent from the night sky, meaning foragers could not attend to a consistent polarisation pattern throughout the night. The moonlight pattern only becomes visible in the sky once the moon reaches –18° below the horizon; thus, waning moon nights present periods when there are neither solar nor lunar cues available to maintain the compass, potentially degrading the compass portion of the vector estimate. This suggests that when the moon is waxing and present throughout the overnight period (at least until moonset), *M. midas* foragers are continuously tracking it and integrating this compass cue into their path integrator. When there are periods overnight when the moon and its polarisation pattern are absent, it is either weighted weakly or the path integrator may become degraded with cues having been interrupted during overnight periods with no detectable celestial cues.

In this vein, it remains unknown if these ants are tracking their lunar polarisation compass by using a time-compensated lunar compass, or if the compass is updated with reference to other cues, such as the panorama, throughout the night. It is possible that these ants form a lunar ephemeris function or time compensator for the moon's position. Solar ephemeris functions are well demonstrated in insect navigators (*Mouritsen and Frost, 2002*; *Massy and Wotton, 2023*), including honeybees (*Dyer and Dickinson, 1994*) as well as desert ants (*Wehner and Lanfranconi, 1981*). This study cannot untangle these possibilities as ants had a continuous view of both the sky and surrounding terrestrial cues throughout the night before testing. However, the shift reductions on nights where there was no access to the lunar polarisation cue until just prior to tests (lunar twilight before moonrise), suggests these foragers may need some set period of exposure to the moon's polarisation pattern to employ it fully during inbound homing. Future work could tackle if a lunar ephemeris function is employed by these ants through exposing or blocking access to the sky and familiar panorama for set time periods when the moon is naturally visible overnight, during waxing gibbous phases.

## Conclusions

Inbound *M. midas* foragers detect and respond predictably to rotations of the moonlight e-vector orientation under a filter and reorient back to the ambient e-vector after filter exit. This ability occurs across the lunar phase, suggesting that polarised moonlight is a detectable cue throughout the lunar month. Heading changes due to polarised moonlight align with responses to polarised sunlight as part of the path integrator during solar twilight. This indicates that polarised moonlight is likely detected and integrated into the ant's path integrator for inbound homing along the same visual pathways as polarised sunlight. Reductions in heading shifts due to differences in PI vector lengths, and periods without access to polarised light patterns suggest that these animals can weight the information provided by celestial polarised light. In so doing these foragers can cater their navigational decisions proportionate to closely match the reliability of available navigational information.

## Materials and methods
### Study site

Experiments were conducted from April through October 2023 on two *Myrmecia midas* nests on the Macquarie University Wallumattagal campus in Sydney, Australia (33°46 11 S, 151°06 40E). *M. midas* nests are typically located within stands of *Eucalyptus* trees with the nest entrance located near (<30 cm) a tree (*Deeti et al., 2024*). *M. midas* is nocturnal, with foraging onset occurring ~20 min after sunset when foragers leave the nest to travel to and up one of several surrounding foraging trees overnight (*Freas et al., 2018*). Inbound navigation is more variable, with foragers returning to the nest entrance overnight and into morning twilight (*Freas et al., 2017b*). For this study, we chose two nests under open canopies to ensure foragers had at least partially unobstructed visual access to

the overhead sky (*Figure 1B*). The understory of these areas is naturally barren of vegetation, and we cleared the foraging column of debris to aid visual tracking.

## Apparatus: Polarisation filter

For each condition, we altered the pattern of polarised moonlight by rotating a linear polarisation filter (Polaroid HN22 analogue, 30 cm diameter) above each ant along their inbound journey. This rotation blocks the ambient e-vector direction of the sky above the navigator, replacing it with an artificial e-vector of polarisation that is rotated ±45° off the ambient pattern. This filter was held by a circular 1 cm thick ring and lifted 10 cm off the ground by four equally spaced thin metal legs (*Figure 1*). All testing was conducted overnight before morning solar twilight, during the lunar twilight for each tested lunar phase. For each night we obtained the moon's position as it reached the horizon based on the Astronomical Almanac (http://asa.usno.navy.mil) and set the ambient lunar e-vector perpendicular to the moon's position at moonset. We relied on a digital compass mobile application (Apple) confirmed by an analogue compass to locate the ambient e-vector and rotated the linear polariser by ±45° from this direction for each ant.

Across all conditions, we recorded four positions: the ant's release point, the position when the filter was placed overhead, the filter exit point, and reorientation after ~1 m, taking care to slowly follow the ant and mark their positions so as to not disturb their travel. These positions determine each forager's initial orientation, exit orientation, and reorientation directions (*Figure 1*). After testing, each forager was marked with acrylic paint (Tamiya) to prevent retesting. Testing was conducted at distinct lunar phases which predictably occur throughout the lunar month cycle (29.5 d).

## Full moon

For full moon testing, we chose nights during which the waxing moon's lunar phase was near full but with clear separation between solar and lunar twilights. Testing on true full moon nights is problematic as solar and lunar twilights fully overlap and the solar polarised light pattern would overpower the lunar counterpart. Testing during the nights preceding the full moon (waxing phase) ensured the moon's presence in the night sky overnight and testing occurred on nights in which the lunar twilight (1 am – 4 am) was clearly separated from the start of morning solar twilight (5:22 am) with illumination above 80% of the lunar surface.

Outbound *M. midas* foragers from two nests were followed as they left the nest during evening twilight and collected as they climbed onto their foraging tree, (Nest 1: 6.0 m; Nest 2: 3.1 m from the nest entra). Each forager was provided a small amount of honey and held within a clear plastic phial on the ground 5 m from the foraging tree with an unobstructed view of the sky (*Figure 1B*). Foragers were held in these phials overnight until the moon's position was within ±10° of the horizon (large stands of trees and buildings near the western horizon occluded the moon's position during all testing; *Figure 1B*).

## Waxing lunar phase

We observed clear shifts in forager headings at both nests under *Full Moon* conditions, yet nights with over 80% lunar illumination only account for nine nights per lunar cycle. To assess if polarised moonlight can be used throughout the lunar month, we tested Nest 1 foragers in three further conditions representing distinct lunar phases: a *Waxing Quarter Moon*, a *Waxing Crescent Moon*, and a *No Moon* control. For the *Quarter Moon* and *Crescent Moon* conditions, we tested ants identically to full moon conditions; for overnight testing, however, the moon has a different temporal period when it is positioned near the horizon (12 am and 9 pm, respectively).

The procedure was slightly modified for *No Moon* testing as we did not test these foragers on the new moon night (the new moon is only present during the day). We hypothesised that foragers with no available ambient polarisation and suddenly presented with an e-vector pattern might fall back on a memory of the morning solar e-vector as many foragers return during morning twilight and this direction remains consistent across nights. In order to test on a night when there were distinct directional differences between the lunar and morning solar e-vectors (Lunar e-vector: 329°; Morning solar e-vector: 7°; Evening Solar e-vector: 353°), we chose to test on a quarter moon night when the moon was well below the horizon. Testing commenced at 9:00 pm, when the moon was 30° below the horizon and we rotated the filter around to the future lunar e-vector direction (moonrise at 12:39 am).

If foragers were relying on a solar e-vector memory, we expected to see unequal shifts between ±45° (smaller shifts in the +45° condition and larger shifts in the −45° condition).

## Waning lunar phases

While there is a consistent presence of polarised light during the waxing phase, during waning lunar phases there is a gap which may impede this pattern's use as a compass cue. Testing during the waxing lunar phase, as the moon's illuminated surface increases, corresponds with periods in which the moon rises prior to sunset and sets overnight. This creates a consistent presence of polarised light (solar then lunar) that foragers could use to continuously update their celestial compass and path integrator. In contrast, the waning lunar phase corresponds with the moon rising overnight, leading to a gap in this cue as solar twilight ends. We hypothesised that the absence of the moon's presence as a cue overnight might decrease its weighting or degrade its integration into the celestial compass. We added two conditions to test this hypothesis: a *Waning Full Moon* and a *Waning Quarter Moon* conditions. Foragers were tested identically to previous conditions; only, they were tested during moonrise (10 pm and 1 am) overnight rather than moonset.

## Vector testing

After noticing differences in heading shift magnitude between nests which correlated with path integration-derived vector lengths, we tested the hypothesis that, similar to solar polarisation (*Freas et al., 2017c*), the navigator's accumulated vector length impacts orientation to rotated lunar polarised light. While foragers at Nest 1 (6.0 m vector) reoriented almost fully, those at Nest 2 (3.1 m vector) only altered their headings by about half of the 45° e-vector shift (25.2° ± 3.7°), despite being tested on near-full-moon nights.

To test this hypothesis, we tested a separate group of foragers at Nest 1 on near (waxing) full moon nights at the same spatial location, but with diverging remaining vectors (*Figure 6A*). In the first test (called *Halfway Collect & Release*), outbound foragers were only allowed to travel half the distance (~3 m) to their foraging tree before being collected and were then released back at this spot to be tested under the ±45° filter rotations at ~2 m from the nest with a small remaining vector (*Figure 6A*). In contrast, the second testing group (*Halfway Release*) was allowed to accumulate their full 6 m vector by collecting outbound foragers as they reached their foraging tree. When these foragers were released, we placed them at the halfway point, 3.0 m from the nest, and tested them with a larger remaining vector near (~2 m) to the nest. We followed outbound foragers to their foraging tree, collected and held them until lunar twilight (2-4 am) identically to previous full moon conditions. In both conditions, we again recorded the initial orientation, filter exit orientation, and post-filter reorientation of each forager. These conditions tested if the vector state (near zero vs. near full) or the test site underlies the observed heading shift differences.

## Statistical analysis

Data were analysed with circular statistics with the statistics package Oriana Version 4 (Kovach Computing Services). Each ant had a slightly different inbound heading due to their stereotypical route along the foraging corridor and we corrected this variance by designating the initial headings (pre-filter) as 0° for calculating shift magnitudes under the filter. To assess shift magnitude between −45° and +45° foragers within conditions, we calculated the mirror of the shifts in each −45° condition, allowing shift magnitude comparisons between conditions within each test. Mirroring the −45° conditions was calculated by mirroring each individual forager's shift across the 0° to 180° plane. This mirrored data set was then compared to the corresponding unaltered +45 condition. As mirrored −45° shifts were not significantly different from +45° shifts in any condition (Watson–Williams F-tests), they were combined for between-condition comparisons. Within-individual comparisons (Initial Orientation vs. Filter Exit and Filter Exit vs. Reorientation) were analysed using Moore's Paired Tests. Across-condition shift magnitudes were analysed using Watson–Williams F-tests. In the lunar phase comparisons where full, quarter, and crescent shift magnitudes, as well as Waning and Waxing phases, were compared, Holm-Bonferroni corrections were applied to the *p*-value to account for multiple comparisons. Given Watson–Williams F-tests can be sensitive to the difference in a spread, when differences between conditions were significant, we further analysed these conditions with Var tests to assess if these differences were due to variance (*Wystrach et al., 2014*; *Freas and Cheng, 2017a*;

*Freas and Spetch, 2019c*). Here, the absolute angular error from the mean vector was calculated for each condition and this error magnitude between conditions was compared using Mann-Whitney u tests.

## Acknowledgements

This project was funded by a Macquarie University Research Fellowship (MQRF0001094), by Macquarie University, and by an ARC Discovery Grant (DP200102337). *Land Acknowledgement*. This work was conducted on the Wallumattagal campus of Macquarie University. We acknowledge the traditional custodians of the land on which Macquarie University sits, the Wallumattagal clan of the Dharug Nation. Their culture and customs have nurtured and sustained this land since the Dreamtime and continue to do so. We pay our respects to their Elders, past, present, and future.

## Additional information

### Funding

| Funder | Grant reference number | Author |
|--------|------------------------|--------|
| Macquarie University | MQRF0001094 | Cody A Freas |
| Australian Research Council | DP200102337 | Ajay Narenda Ken Cheng |

The funders had no role in study design, data collection and interpretation, or the decision to submit the work for publication.

### Author contributions

Cody A Freas, Conceptualization, Data curation, Formal analysis, Funding acquisition, Validation, Investigation, Visualization, Methodology, Writing - original draft, Project administration, Writing – review and editing; Ajay Narenda, Conceptualization, Funding acquisition, Project administration, Writing – review and editing; Trevor Murray, Conceptualization, Methodology, Writing – review and editing; Ken Cheng, Conceptualization, Funding acquisition, Methodology, Project administration, Writing – review and editing

### Author ORCIDs

Cody A Freas ⓘ https://orcid.org/0000-0001-7026-1255

Reviewer #1 (Public review): https://doi.org/10.7554/eLife.97615.4.sa1
Reviewer #2 (Public review): https://doi.org/10.7554/eLife.97615.4.sa2
Reviewer #3 (Public review): https://doi.org/10.7554/eLife.97615.4.sa3
Author response https://doi.org/10.7554/eLife.97615.4.sa4

## Additional files

### Supplementary files
• MDAR checklist

### Data availability

All data, documentation and code is made available online at osf.io (https://doi.org/10.17605/OSF.IO/J8P2N).

The following dataset was generated:

| Author(s) | Year | Dataset title | Dataset URL | Database and Identifier |
|-----------|------|---------------|-------------|-------------------------|
| Freas CA | 2024 | Lunar Polarisation Cue Data | https://osf.io/j8p2n/ | Open Science Framework, 10.17605/OSF.IO/J8P2N |

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
