## [Editor Report · eLife Assessment]

This **important** study substantially advances our understanding of nocturnal animal navigation and the ways that animals use polarized light. The evidence supporting the conclusions is **convincing**, with elegant behavioural experiments in actively navigating ants. The work will be of interest to biologists working on animal navigation or sensory ecology.

---

## [Referee Report · Reviewer #1 (Public review)]

Freas et al. investigated if the exceedingly dim polarization pattern produced by the moon can be used by animal to guide a genuine navigational task. The sun and moon are celestial beacons for directional information, but they can be obscured by clouds, canopy, or the horizon. However, even when hidden from view, these celestial bodies provide directional information through the polarized light patterns in the sky. While the sun's polarization pattern is famously used by many animals for compass orientation, until now it has never been shown that the extremely dim polarization pattern of the moon can be used for navigation. To test this, Freas et al. studied nocturnal bull ants, by placing a linear polarizer in the homing path on a freely navigating ant 45 degrees shifted to the moon's natural polarization pattern. They recorded the homing direction of an ant before entering the polarizer, under the polarizer, and again after leaving the area covered by the polarizer. The results very clearly show, that ants walking under the linear polarizer change their homing direction by about 45 degrees in comparison to the homing direction under the natural polarization pattern and change it back after leaving the area covered by the polarizer again. These results can be repeated throughout the lunar month, showing that bull ants can use the moon's polarization pattern even under crescent moon conditions. Finally, the authors show, that the degree in which the ants change their homing direction is dependent on the length of their home vector, just as it is for the solar polarization pattern.

The behavioral experiments are very well designed, and the statistical analyses are appropriate for the data presented. The authors' conclusions are nicely supported by the data and clearly show nocturnal bull ants use the dim polarization pattern of the moon for homing, in the same way many animals use the sun's polarization pattern during the day. This is the first proof of the use of the lunar polarization pattern in any animal.

Comments on revised version:

The authors have addressed all of my previous comments and suggestions. I am happy with the way the manuscript has improved and have no further comments.

---

## [Referee Report · Reviewer #2 (Public review)]

Summary:

The authors aimed to understand whether polarised moonlight could be used as a directional cue for nocturnal animals homing at night, particularly at times of night when polarised light is not available from the sun. To do this, the authors used nocturnal ants, and previously established methods, to show that the walking paths of ants can be altered predictably when the angle of polarised moonlight illuminating them from above is turned by a known angle (here +/- 45 degrees).

Strengths:

The behavioural data are very clear and unambiguous. The results clearly show that when the angle of downwelling polarised moonlight is turned, ants turn in the same direction. The data also clearly show that this result is maintained even for different phases (and intensities) of the moon, although during the waning cycle of the moon the ants' turn is considerably less than may be expected.

Impact:

The authors have discovered that nocturnal bull ants, while homing back to their nest holes at night, are able to use the dim polarised light pattern formed around the moon for path integration. Even though similar methods have previously shown the ability of dung beetles to orient along straight trajectories for short distances using polarised moonlight, this the first evidence of an animal that uses polarised moonlight in homing. This is quite significant, and their findings are well supported by their data.

Comments on revised version:

The authors have made a good effort to accommodate my suggestions for improvement (and from what I can tell, those of the other reviewers). I have no further comments.

---

## [Referee Report · Reviewer #3 (Public review)]

Summary:

This manuscript presents a series of experiments aimed at investigating orientation to polarized lunar skylight in a nocturnal ant, the first report of its kind that I am aware of.

Strengths:

The study was conducted carefully and is clearly explained here.

Comments on revised version:

The manuscript is much improved and will make an excellent contribution to the field.

---

## [Author Response]

The following is the authors’ response to the previous reviews.

**Public Reviews:**

**Reviewer #1 (Public Review):**
Freas et al. investigated if the exceedingly dim polarization pattern produced by the moon can be used by animals to guide a genuine navigational task. The sun and moon have long been celestial beacons for directional information, but they can be obscured by clouds, canopy, or the horizon. However, even when hidden from view, these celestial bodies provide directional information through the polarized light patterns in the sky. While the sun's polarization pattern is famously used by many animals for compass orientation, until now it has never been shown that the extremely dim polarization pattern of the moon can be used for navigation. To test this, Freas et al. studied nocturnal bull ants, by placing a linear polarizer in the homing path on freely navigating ants 45 degrees shifted to the moon's natural polarization pattern. They recorded the homing direction of an ant before entering the polarizer, under the polarizer, and again after leaving the area covered by the polarizer. The results very clearly show, that ants walking under the linear polarizer change their homing direction by about 45 degrees in comparison to the homing direction under the natural polarization pattern and change it back after leaving the area covered by the polarizer again. These results can be repeated throughout the lunar month, showing that bull ants can use the moon's polarization pattern even under crescent moon conditions. Finally, the authors show, that the degree in which the ants change their homing direction is dependent on the length of their home vector, just as it is for the solar polarization pattern.The behavioral experiments are very well designed, and the statistical analyses are appropriate for the data presented. The authors' conclusions are nicely supported by the data and clearly show that nocturnal bull ants use the dim polarization pattern of the moon for homing, in the same way many animals use the sun's polarization pattern during the day. This is the first proof of the use of the lunar polarization pattern in any animal.
**Reviewer #2 (Public Review):**
Summary:The authors aimed to understand whether polarised moonlight could be used as a directional cue for nocturnal animals homing at night, particularly at times of night when polarised light is not available from the sun. To do this, the authors used nocturnal ants, and previously established methods, to show that the walking paths of ants can be altered predictably when the angle of polarised moonlight illuminating them from above is turned by a known angle (here +/- 45 degrees).Strengths:The behavioural data are very clear and unambiguous. The results clearly show that when the angle of downwelling polarised moonlight is turned, ants turn in the same direction. The data also clearly show that this result is maintained even for different phases (and intensities) of the moon, although during the waning cycle of the moon the ants' turn is considerably less than may be expected.Weaknesses:The final section of the results - concerning the weighting of polarised light cues into the path integrator - lacks clarity and should be reworked and expanded in both the Methods and the Results (also possibly with an extra methods figure). I was really unsure of what these experiments were trying to show or what the meaning of the results actually are.

Rewrote these sections and added figure panel to Figure 6.

Impact:The authors have discovered that nocturnal bull ants while homing back to their nest holes at night, are able to use the dim polarised light pattern formed around the moon for path integration. Even though similar methods have previously shown the ability of dung beetles to orient along straight trajectories for short distances using polarised moonlight, this is the first evidence of an animal that uses polarised moonlight in homing. This is quite significant, and their findings are well supported by their data.
**Reviewer #3 (Public Review):**
Summary:This manuscript presents a series of experiments aimed at investigating orientation to polarized lunar skylight in a nocturnal ant, the first report of its kind that I am aware of.Strengths:The study was conducted carefully and is clearly explained here.Weaknesses:I have only a few comments and suggestions, that I hope will make the manuscript clearer and easier to understand.Time compensation or periodic snapshotsIn the introduction, the authors compare their discovery with that in dung beetles, which have only been observed to use lunar skylight to hold their course, not to travel to a specific location as the ants must. It is not entirely clear from the discussion whether the authors are suggesting that the ants navigate home by using a time-compensated lunar compass, or that they update their polarization compass with reference to other cues as the pattern of lunar skylight gradually shifts over the course of the night - though in the discussion they appear to lean towards the latter without addressing the former. Any clues in this direction might help us understand how ants adapted to navigate using solar skylight polarization might adapt use to lunar skylight polarization and account for its different schedule. I would guess that the waxing and waning moon data can be interpreted to this effect.

Added a paragraph discussing this distinction in mechanisms and the limits of the current data set in untangling them. An interesting topic for a follow up to be sure.

Effects of moon fullness and phase on precisionAs well as the noted effect on shift magnitudes, the distributions of exit headings and reorientations also appear to differ in their precision (i.e., mean vector length) across moon phases, with somewhat shorter vectors for smaller fractions of the moon illuminated. Although these distributions are a composite of the two distributions of angles subtracted from one another to obtain these turn angles, the precision of the resulting distribution should be proportional to the original distributions. It would be interesting to know whether these differences result from poorer overall orientation precision, or more variability in reorientation, on quarter moon and crescent moon nights, and to what extent this might be attributed to sky brightness or degree of polarization.

See below for response to this and the next reviewer comment

N.B. The Watson-Williams tests for difference in mean angle are also sensitive to differences in sample variance. This can be ruled out with another variety of the test, also proposed by Watson and Williams, to check for unequal variances, for which the F statistic is = (n2-1)*(n1-R1) / (n1-1)*(n2-R2) or its inverse, whichever is >1.

We have looked at the amount of variance from the mean heading direction in terms of both the shifts and the reorientations and found no significant difference in variance between all relevant conditions. It is possible (and probably likely) that with a higher n we might find these differences but with the current data set we cannot make statistical statements regarding degradations in navigational precision.

As an additional analysis to address the Watson-Williams test‘s sensitivity to changes in variance, we have added var test comparisons for each of the comparisons, which is a well-established test to compare variance changes. None of these were significantly different, suggesting the observed differences in the WW tests are due to changes in the mean vector and not the distribution. We have added this test to the text.

**Recommendations for the authors:**

**Reviewer #1 (Recommendations For The Authors):**
I have only very few minor suggestions to improve the manuscript:(1) While I fully agree with the authors that their study, to the best of my knowledge, provides the first proof (in any animal) of the use of the moon's polarization pattern, the many repetitions of this fact disturb the flow of the text and could be cut at several instances.

Yes, it is indeed repeated to an annoying degree.

We have removed these beyond bookending mentions (Abstract and Discussion).

(2) In my opinion, the authors did not change the "ambient polarization pattern" when using the linear polarization filter (e.g., l. 55, 170, 177 ...). The linear polarizer presents an artificial polarization pattern with a much higher degree of polarization in comparison to the ambient polarization pattern. I would suggest re-phrasing this, to emphasize the artificial nature of the polarization pattern under the polarizer.

We have made these suggested changes throughout the text to clarify. We no longer say the ambient pattern was

(3) Line 377: I do not see the link between the sentence and Figure 7

Changed where in the discussion we refer to Figure 7.

(4) Figure 7 upper part: In my opinion, the upper part of Figure 7 does not add any additional value to the illustration of the data as compared to Figure 5 and could be cut.

We thought it might be easier for some reader to see the shifts as a dial representation with the shift magnitude converted to 0-100% rather than the shifts in Figure 5. This makes it somewhat like a graphical abstract summarising the whole study.

I agree that Figure 5 tells the same story but a reader that has little background in directional stats might find figure 7 more intuitive. This was the intent at least.

If it becomes a sticking point, then we can remove the upper portion.

**Reviewer #2 (Recommendations For The Authors):**
MINOR CORRECTIONS AND QUERIESLine 117: THE majority

Corrected

Lines 129-130: Do you have a reference to support this statement? I am unaware of experiments that show that homing ants count their steps, but I could have missed it.

We have added the references that unpack the ant pedometer.

Line 140: remove "the" in this line.

Removed

Line 170: We need more details here about the spectral transmission properties of the polariser (and indeed which brand of filter, etc.). For instance, does it allow the transmission of UV light?

Added

Line 239: "...tested identicALLY to ...."

Corrected

Lines 242-258 (Vector testing): I must admit I found the description of these experiments very difficult to follow. I read this section several times and felt no wiser as a result. I think some thought needs to be given to better introduce the reader to the rationale behind the experiment (e.g., start by expanding lines 243-246, and maybe add a methods figure that shows the different experimental procedures).

I have rewritten this section of the methods to clearly state the experiment rational and to be clearer as to the methodology.

Also Added a methods panel to Figure 6.

Line 247: "reoriented only halfway". What does this mean? Do you mean with half the expected angle?

Yes, this is a bit unclear. We have altered for clarity:

‘only altered their headings by about half of the 45° e-vector shift (25.2°± 3.7°), despite being tested on near-full-moon nights.’

Results section (in general): In Figure 1 (which is a very nice figure!) you go to all the trouble of defining b degrees (exit headings) and c degrees (reorientation headings), which are very intuitive for interpreting the results, and then you totally abandon these convenient angles in favour of an amorphous Greek symbol Phi (Figs. 2-6) to describe BOTH exit and reorientation headings. Why?? It becomes even more confusing when headings described by Phi can be typically greater than 300 degrees in the figures, but they are never even close to this in the text (where you seem to have gone back to using the b degrees and c degrees angles, without explicitly saying so). Personally, I think the b degrees and c degrees angles are more intuitive (and should be used in both the text and the figures), but if you do insist on using Phi then you should use it consistently in both the text and the figures.

Replaced Phi with b° and c° for both figures and in the text.

Finally, for reorientation angles in Figure 4A, you say that the angle is 16.5 degrees. This angle should have been 143.5 degrees to be consistent with other figures.

Yes, the reorientation was erroneously copied from the shift data (it is identical in both the +45 shift and reorientation for Figure 4A). This has now been corrected

Line 280, and many other lines: Wherever you refer to two panels of the same figure, they should be written as (say) Figure 2A, B not Figure 2AB.

Changed as requested throughout the text.

Line 295 (Waxing lunar phases): For these experiments, which nest are you using? 1 or 2?

We have added that this is nest 1.

Figure 3B: The title of this panel should be "Waxing Crescent Moon" I think.

Ah yes, this is incorrect in the original submission. I have fixed this.

Lines 312-313: Here it sounds as though the ants went right back to the full +/- 45 degrees orientations when they clearly didn't (it was -26.6 degrees and 189.9 degrees). Maybe tone the language down a bit here.

Changed this to make clear the orientation shift is only ‘towards’ the ambient lunar e-vector.

Line 327: Insert "see" before "Figure 5"

Added

Line 329: See comment for Line 295.

We have added that this is nest 1.

Lines 357-373 (Vector testing): Again, because of the somewhat confusing methods section describing these experiments, these results were hard to follow, both here and in the Discussion. I don't really understand what you have shown here. Re-think how you present this (and maybe re-working the Methods will be half the battle won).

I have rewritten these sections to try to make clear these are ant tested with differences in vector length 6m vs. 2m, tested at the same location. Hopefully this is much clearer, but I think if these portions remain a bit confusing that a full rename of the conditions is in order. Something like long vector and short vector would help but comes with the problem of not truly describing what the purpose of the test is which is to control for location, thus the current condition names. As it stands, I hope the new clarifications adequately describe the reasoning while keeping the condition names. Of course, I am happy to make more changes here as making this clear to readers is important for driving home that the path integrator is in play.

See current change to results as an example: ‘Both forgers with a long ~6m remaining vector (*Halfway Release),* or a short ~2m remaining vector (*Halfway Collection & Release),* tested at the same location_,_ exhibited significant shifts to the right of initial headings when the e-vector was rotated clockwise +45°.’

Line 361: I think this should be 16.8 not 6.8

Yes, you are correct. Fixed in text (16.8).

Line 365: I think this should be -12.7 not 12.7

Yes, you are correct. Fixed in text (–12.7).

Line 408: "morning twilight". Should this be "morning solar twilight"? Plus "M midas" should be "M. midas"

Added and fixed, respectively.

Line 440. "location" is spelt wrong.

Fixed spelling.

Line 444: "...WITH longer accumulated vectors, ..."

Added ‘with’ to sentence.

Line 447: Remove "that just as"

Removed.

Line 448: "Moonlight polarised light" should be "Polarised moonlight"

Corrected.

Lines 450-453: This sentence makes little sense scientifically or grammatically. A "limiting factor" can't be "accomplished". Please rephrase and explain in more detail.

This sentence has been rephrased:

‘The limiting factors to lunar cue use for navigation would instead be the ant’s detection threshold to either absolute light intensity, polarization sensitivity and spectral sensitivity. Moonlight is less UV rich compared to direct sunlight and the spectrum changes across the lunar cycle (Palmer and Johnsen 2015).’

Line 474: Re-write as "... due to the incorporation of the celestial compass into the path integrator..."

Added.

**Reviewer #3 (Recommendations For The Authors):**
Minor commentsLine 84 I am not sure that we can infer attentional processes in orientation to lunar skylight, at least it has not yet been investigated.

Yes, this is a good point. We have changed ‘attend’ to ‘use’.

Line 90 This description of polarized light is a little vague; what is meant by the phrase "waves which occur along a single plane"? (What about the magnetic component? These waves can be redirected, are they then still polarized? Circular polarization?). I would recommend looking at how polarized light is described in textbooks on optics.

Response: We have rewritten the polarised light section to be clearer using optics and light physics for background.

Line 92 The phrase "e-vector" has not been described or introduced up to this point.

We now introduce e-vector and define it.

‘Polarised light comprises light waves which occur along a single plane and are produced as a by-product of light passing through the upper atmosphere (Horváth & Varjú 2004; Horváth et al., 2014). The scattering of this light creates an e-vector pattern in the sky, which is arranged in concentric circles around the sun or moon's position with the maximum degree of polarisation located 90° from the source. Hence when the sun/moon is near the horizon, the pattern of polarised skylight is particularly simple with uniform direction of polarisation approximately parallel to the north-south axes (Dacke et al., 1999, 2003; Reid et al. 2011; Zeil et al., 2014).’

Happy to make further changes as well.

Line 107 Diurnal dung beetles can also orient to lunar skylight if roused at night (Smolka et al., 2016), provided the sky is bright enough. Perhaps diurnal ants might do the same?

Added the diurnal dung beetles mention as well as the reference.

Also, a very good suggestion using diurnal bull ants.

Line 146 Instead of lunar calendar the authors appear to mean "lunar cycle".

Changed

Line 165 In Figure 1B, it looks like visual access to the sky was only partly "unobstructed". Indeed foliage covers as least part of the sky right up to the zenith.

We have added that the sky is partially obstructed.

Line 179 This could also presumably be checked with a camera?

For this testing we tried to keep equipment to a minimum for a single researcher walking to and from the field site given the lack of public transport between 1 and 4am. But yes, for future work a camera based confirmation system would be easier.

Line 243 The abbreviation "PI" has not been described or introduced up to this point.

Changes to ‘path integration derived vector lengths….’

Line 267 The method for comparing the leftwards and rightwards shifts should be described in full here (presumably one set of shifts was mirrored onto the other?).

We have added the below description to indicate the full description of the mirroring done to counterclockwise shifts.

‘To assess shift magnitude between −45° and +45° foragers within conditions, we calculated the mirror of shift in each −45° condition, allowing shift magnitude comparisons within each condition. Mirroring the −45° conditions was calculated by mirroring each shift across the 0° to 180° plane and was then compared to the corresponding unaltered +45 condition.’

Discussion Might the brightness and spectrum of lunar skylight also play a role here?

We have added a section to the discussion to mention the aspects of moonlight which may be important to these animals, including the spectrum, brightness and polarisation intensity.

Line 451 The sensitivity threshold to absolute light intensity would not be the only limiting factor here. Polarization sensitivity and spectral sensitivity may also play a role (moonlight is less UV rich than sunlight and the spectrum of twilight changes across the lunar cycle: Palmer & Johnsen, 2015).

Added this clarification.

Line 478 Instead of the "masculine ordinal" symbol used (U+006F) here a degree symbol (U+00B0) should be used.

Ah thank you, we have replaced this everywhere in the text.

Line 485 It should be possible to calculate the misalignment between polarization pattern before and after this interruption of celestial cues. Does the magnitude of this misalignment help predict the size of the reorientation?

Reorientations are highly correlated with the shift size under the filter, which makes sense as larger shifts mean that foragers need to turn back more to reorient to both the ambient pattern and to return to their visual route. Reorientation sizes do not show a consistent reduction compared to under-the-filter shifts when the lunar phase is low and is potentially harder to detect.

I have reworked this line in the text as I do not think there is much evidence for misalignment and it might be more precise to say that overnight periods where the moon is not visible may adversely impact the path integrator estimate, though it is currently unknown the full impact of this celestial cue gap of if other cues might also play a role.

Line 642 "from their" should be "relative to"

Changed as requested

Figure 1B Some mention should be made of the differences in vegetation density.

Added a sentence to the figure caption discussing the differences in both vegetation along the horizon and canopy cover.

Figures 2-6 A reference line at 0 degrees change might help the reader to assess the size of orientation changes visually. Confidence intervals around the mean orientation change would also help here.

We have now added circular grid lines and confidence intervals to the circular plots. These should help make the heading changes clear to readers.